# 4DNeX: Feed-Forward 4D Generative Modeling Made Easy

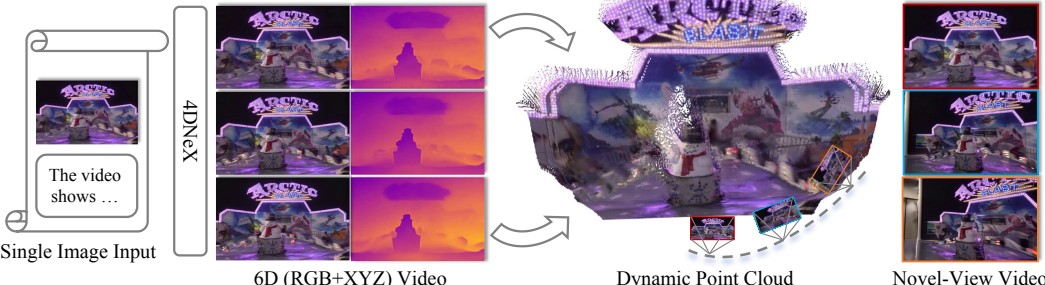

Figure 1: **4DNeX generates 6D video from a single image to enable 4D scene creation and novel-view video rendering.**

## Abstract

We present **4DNeX**, the first feed-forward framework for generating 4D (*i.e.*, dynamic 3D) scene representations from a single image. In contrast to existing methods that rely on computationally intensive optimization or require multi-frame video inputs, 4DNeX enables efficient, end-to-end image-to-4D generation by fine-tuning a pretrained video diffusion model. Specifically, **1)** to alleviate the scarcity of 4D data, we construct 4DNeX-10M, a large-scale dataset with high-quality 4D annotations generated using advanced reconstruction approaches. **2)** we introduce a unified 6D video representation that jointly models RGB and XYZ sequences, facilitating structured learning of both appearance and geometry. **3)** we propose a set of simple yet effective adaptation strategies to repurpose pretrained video diffusion models for 4D modeling. 4DNeX produces high-quality dynamic point clouds that enable novel-view video synthesis. Extensive experiments demonstrate that 4DNeX outperforms existing 4D generation methods in efficiency and generalizability, offering a scalable solution for image-to-4D modeling and laying the foundation for generative 4D world models that simulate dynamic scene evolution.

## 1 Introduction

The images we capture are 2D projections of the 4D (*i.e.*, dynamic 3D) physical world. Creating a 4D scene from such 2D observations, particularly from a single image, is a highly challenging yet compelling task. As a core capability in generative modeling, image-to-4D generation lays the foundation for building 4D world models that can predict and simulate dynamic scene evolution, enabling a wide range of applications in AR/VR, film production, and digital content creation.

Existing approaches for 4D scene modeling can be broadly classified into two categories. The first comprises 4D generation methods, which typically adopt representations such as Neural Radiance Fields (NeRF) Mildenhall et al. (2022) or 3D Gaussian Splatting (3DGS) Kerbl et al. (2023). These methods can be further divided into feed-forward Ren et al. (2025a); Wu et al. (2024b); Zhao et al. (2024); Sun et al. (2024b) and optimization-based variants Liu et al. (2025); Yu et al. (2024a); Zheng et al. (2024); Bahmani et al. (2024); Zhao et al. (2023); Ren et al. (2023). However, they either require video input or rely on object-centric, computationally intensive optimization procedures. The second category includes dynamic Structure-from-Motion (SfM) approaches Li et al. (2024); Zhang

et al. (2024a); Xu et al. (2025); Jiang et al. (2025); Wang et al. (2025b), which estimate dynamic 3D structures such as time-varying point clouds from video sequences. However, these methods remain incapable of generating 4D representations from a single image.

To this end, we aim to develop a feed-forward framework for 4D scene generation from a single image. A straightforward solution is to fine-tune a pretrained video diffusion model. However, this approach presents two core challenges: **1)** how to mitigate the scarcity of 4D data, and **2)** how to adapt the pretrained model in a simple and efficient way.

For the **first** challenge, we curate 4DNeX-10M, a large-scale dataset comprising both static and dynamic scenes, with high-quality 4D annotations generated from monocular videos using state-of-the-art reconstruction methods Wang et al. (2024; 2025a;c); Zhang et al. (2024a); Li et al. (2024). To ensure geometric accuracy and scene diversity, we apply careful data selection, pseudo-annotation generation, and multi-stage filtering. To address the **second** challenge, we first introduce a unified 6D video representation that models RGB and XYZ sequences jointly, enabling the structured modeling of both appearance and geometry. We then systematically investigate different fusion strategies between the two modalities and show that width-wise fusion achieves the most effective cross-modal alignment. Moreover, we incorporate a set of carefully designed techniques, including XYZ initialization, XYZ normalization, mask design, and modality-aware token encoding, to adapt pretrained video diffusion models in a simple manner while preserving their generative priors.

To summarize, we present 4DNeX, the first feed-forward framework for image-to-4D generation (Fig. 1). We qualitatively demonstrate the plausibility of the generated dynamic point clouds. Furthermore, to validate their utility, we leverage TrajectoryCrafter YU et al. (2025) to transform the generated 4D point clouds into novel-view videos, achieving comparable results to existing 4D generation methods. In addition, we perform comprehensive ablation studies to validate the effectiveness of our proposed fine-tuning strategies.

Our main contributions are as follows. First, we propose 4DNeX, the first feed-forward framework for image-to-4D generation, capable of producing dynamic point clouds from a single image. Second, we construct 4DNeX-10M, a large-scale dataset with high-quality 4D annotations. Finally, we introduce a set of effective fine-tuning strategies to adapt pretrained video diffusion models for 4D generation.

## 2    RELATED WORK

**Optimization-based 4D Generation.** Recent works leverage pre-trained diffusion priors Ho & Salimans (2022); Song et al. (2020); Ho et al. (2020) to optimize 3D/4D representations Mildenhall et al. (2022); Pumarola et al. (2021); Kerbl et al. (2023); Wu et al. (2024a) with synthesized multi-view images or score distillation Poole et al. (2022). A key challenge is ensuring spatial–temporal consistency of the guidance. Some methods Yu et al. (2024a); Rahamim et al. (2024); Zheng et al. (2024); Zhao et al. (2023); Gao et al. (2024a); Bahmani et al. (2024); Jiang et al. (2023); Zeng et al. (2024) extend static-image 3D representations with dynamic cues from video diffusion models, while others Sun et al. (2024a); Pan et al. (2024); Yin et al. (2023); Singer et al. (2023b); Ren et al. (2023); Liu et al. (2025) start from video generation to enforce cross-view consistency. The latest Free4D Liu et al. (2025) generates multi-view videos in a training-free manner with consistency-preserving designs, but is limited to small camera and scene motion. Beyond consistency, optimization-based methods suffer from high cost, long runtime, and instability from multi-stage optimization. In this work, we propose a feed-forward 4D framework as a more efficient and scalable alternative.

**Feed-forward 4D Generation.** These methods directly predict 4D representations in a single pass, avoiding the cost and inconsistency of optimization-based pipelines, and enabling efficient end-to-end learning of spatiotemporal structures. Some focus on generating temporally consistent, viewpoint-controllable videos Zhao et al. (2024); Sun et al. (2024b); Ren et al. (2025b); DeepMind (2025): GenXD Zhao et al. (2024) fuses camera and image conditions but still requires post-optimization for explicit 4D geometry, while DimensionX Sun et al. (2024b) introduces motion-specific LoRA for dynamic view synthesis but lacks fully free-view control. Other works aim to directly generate 4D. L4GM Ren et al. (2025a) extends LGM Tang et al. (2024) with per-frame Gaussian splats and temporal self-attention, and Cat4D Wu et al. (2024b) finetunes CAT3D Gao et al. (2024b) on pseudo-4D data but struggles to generalize. TesserAct Zhen et al. (2025) predicts RGB, depth, normals, and motion from a single image for embodied robotics, but its multitask design is task-

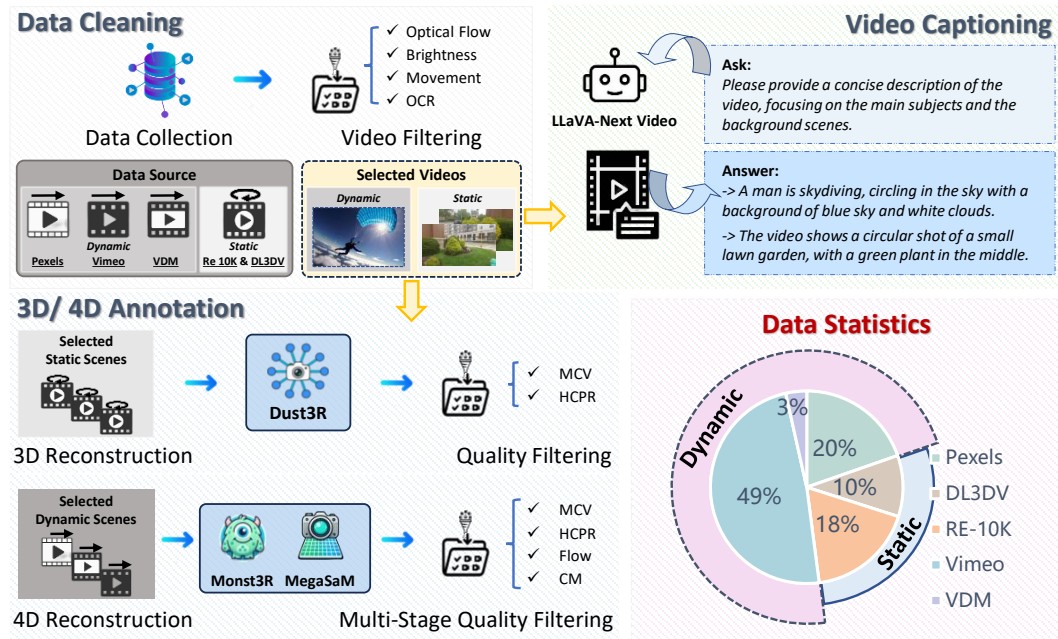

Figure 2: **Data Curation Pipeline.** The video data is collected from various sources and then selected by video filtering during Data Cleaning. The selected data is captioned via LLaVA-Next-Video model in Video Captioning. The selected data is processed and finally filtered out the video with high-quality annotation during 3D/4D Annotation. Data statistics is also provided in bottom right.

specific and not suited for in-the-wild scenarios. In contrast, we efficiently generate general-purpose 4D representations from a single image by leveraging pre-trained video diffusion models and a transferable training paradigm. Related dynamic SfM methods Li et al. (2024); Zhang et al. (2024a); Xu et al. (2025); Jiang et al. (2025); Wang et al. (2025b); Huang et al. (2025) recover time-varying 3D from videos but cannot generate 4D from a single image, focusing on reconstruction rather than generation of appearance–geometry sequences.

**Video Generation Models.** Pre-trained video generation models Ho et al. (2022b;a); Team (2024); Singer et al. (2023a) show strong generative ability and support many downstream tasks. CogVideo Hong et al. (2023) and CogVideoX Yang et al. (2024) adopt expert transformers and 3D full attention for high-quality text-to-video synthesis, while DynamiCrafter Xing et al. (2024) animates input images at arbitrary positions. Beyond text-to-video, recent works target viewpoint-conditioned generation. SynCamMaster Bai et al. (2024) and Collaborative Video Diffusion Kuang et al. (2024) encode camera viewpoints and synchronize multi-view videos. Other approaches bring video generation into the 3D domain: ViewCrafter Yu et al. (2024b) refines lossy multi-view reconstructions, and TrajectoryCrafter YU et al. (2025) builds data pipelines for novel-view dynamic scenes. We leverage TrajectoryCrafter YU et al. (2025) to render generated 4D point clouds into novel-view videos.

## 3 4DNEX-10M

To mitigate data scarcity in 4D generative modeling, we construct 4DNeX-10M, a large-scale hybrid dataset for training feed-forward 4D models. It aggregates public and internal videos of static and dynamic scenes, with rigorous filtering, pseudo-annotation, and quality checks to ensure geometric consistency, motion diversity, and realism. As shown in Fig. A (Appendix), it covers diverse environments (indoor/outdoor, landscapes, close-range, high-speed), scene types (static and human-involved), lighting conditions, and human activities, with precise 4D point maps and camera trajectories. In total, 4DNeX-10M provides over 9.2M annotated video frames. For data curation, as illustrated in Figure 2, we adopt an automated pipeline with three stages: data cleaning, data captioning, and 3D/4D annotation.

## 3.1 DATA PREPROCESSING

The foundation of 4DNeX-10M is built upon a variety of datasets, each contributing distinct scene characteristics and motion types.

**Data Sources.** We collect monocular videos from multiple sources: DL3DV-10K (DL3DV) Ling et al. (2024) and RealEstate10K (RE10K) Zhou et al. (2018) provide static indoor/outdoor scenes with diverse trajectories; Pexels offers large-scale human-centric videos with metadata (motion, OCR, optical flow); Vimeo (from Vchitect 2.0 Fan et al. (2025)) contributes in-the-wild dynamics; and Huang et al. (2024) supplies synthetic dynamic sequences via video diffusion models (VDM).

**Initial Filtering.** For large-scale sources, we use metadata (optical flow, motion, OCR) to discard videos with motion blur or excessive text. For all sources, we further apply brightness filtering based on average luminance ($0.299R + 0.587G + 0.114B$) to remove extreme illumination cases.

**Video Captioning.** For datasets without textual annotations (*e.g.*, DL3DV-10K, RE10K), we use LLaVA-Next-Video Zhang et al. (2024b) to generate captions. We uniformly sample 32 frames per video (or clip) and query the LLaVA-Next-Video-7B-Qwen2 model with a concise-description prompt. For scenes with consistent content (*e.g.*, DL3DV-10K, Dynamic Replica) we generate one caption per video, while RE10K is split into clips for separate captions.

## 3.2 STATIC DATA PROCESSING

To learn strong geometric priors, we use static monocular videos from Ling et al. (2024) and Zhou et al. (2018), which cover diverse environments with varied trajectories for rich multi-view coverage.

**Pseudo 3D Annotation.** Since these datasets lack 3D ground truth, we apply DUSt3R Wang et al. (2024) to recover a consistent scene-level 3D structure.

**Quality Filtering.** To ensure high-quality annotations, we define two metrics using the confidence maps from DUSt3R: 1) the *Mean Confidence Value (MCV)*, averaging pixel-wise confidence scores over all frames, and 2) the *High-Confidence Pixel Ratio (HCPR)*, representing the proportion of pixels exceeding a threshold $\tau$. We select the top-$r\%$ of clips for each metric and retain over 100K high-quality 28-frame clips with reliable pseudo point map annotations for static training.

## 3.3 DYNAMIC DATA PROCESSING

To enrich 4DNeX-10M with dynamic content, we collect monocular videos from Pexels, VDM, and Vimeo. These datasets contain diverse real-world scenes with motion and depth variation but lack ground-truth geometry.

**Pseudo 4D Annotation.** We employ MonST3R Zhang et al. (2024a) and MegaSaM Li et al. (2024), two advanced dynamic reconstruction models, to generate pseudo 4D annotations. Each model recovers temporally coherent 3D point clouds and globally aligned camera poses from monocular videos, enabling the construction of time-varying scene representations.

**Multi-Stage Filtering.** To select high-quality clips, we apply three sequential filtering strategies. First, we use the final alignment loss in the global fusion stage, which reflects multi-view consistency and flow agreement with RAFT Teed & Deng (2021), to filter out low-quality reconstructions. Second, we assess camera smoothness (CS) by computing frame-wise velocity and acceleration from camera translations, and estimate local trajectory curvature as:

$$\kappa_i = \frac{\|\mathbf{v}_{i+1} - \mathbf{v}_i\|}{\|\mathbf{v}_{i+1}\|^2 + \|\mathbf{v}_i\|^2 + \epsilon}, \quad \epsilon > 0. \tag{1}$$

Clips with low average velocity, acceleration, and curvature are retained. Third, we apply the same *Mean Confidence Value (MCV)* and *High-Confidence Pixel Ratio (HCPR)* used in the static pipeline. After filtering, we retain approximately 32K clips from the MonST3R-processed set, 5K clips from VDM, and 27K from Pexels, and over 80k clips from MegaSaM-processed set. Together, these yield a total over 110K high-quality clips with pseudo 4D annotations, enabling robust modeling of dynamic 3D scenes across a wide range of motions and appearances.

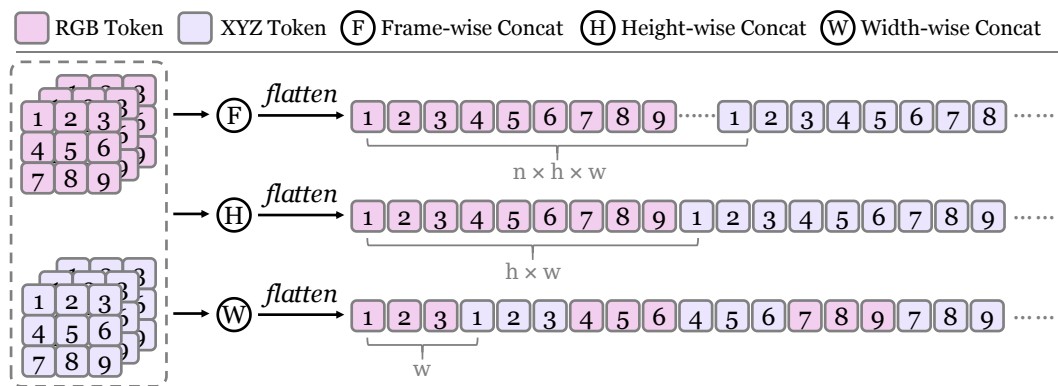

Figure 3: **Comparison of spatial fusion strategies.** We compare frame-, height-, and width-wise fusion in terms of the interaction distance between RGB and XYZ tokens.

# 4  4DNEX

## 4.1  PROBLEM FORMULATION

Given a single image $I_0 \in \mathbb{R}^{H \times W \times 3}$, we aim to construct a 4D (*i.e.*, dynamic 3D) representation of the underlying scene geometry. This task can be formulated as learning a conditional distribution over a sequence of dynamic point clouds:

$$p\left(\{P_t\}_{t=0}^{T-1} \mid I_0\right),\qquad(2)$$

where $\{P_t\}_{t=0}^{T-1}$ denotes the sequence of dynamic point clouds. However, directly modeling point clouds is challenging due to their highly unstructured nature. To address this, inspired by Zhang et al. (2025), we adopt a pixel-aligned point map representation, *XYZ*, where each frame $X_t^{\text{XYZ}} \in \mathbb{R}^{H \times W \times 3}$ encodes the 3D coordinates of each pixel in the global coordinates. This format provides a structured and learnable structure, making it compatible with existing generative models. Instead of directly modeling $\{P_t\}$, we reformulate the problem as predicting paired RGB and XYZ image sequences:

$$p\left(\{X_t^{RGB},\ X_t^{XYZ}\}_{t=0}^{T-1} \mid I_0\right).\qquad(3)$$

Therefore, a 4D scene can be effectively represented using a 6D video composed of paired RGB and XYZ sequences. This simple and unified representation offers two key advantages: it enables explicit 3D consistency supervision through pixel-aligned XYZ maps, and eliminates the need for camera control, facilitating scalable and robust 4D generation.

To model this distribution, we adopt Wan2.1 Wan et al. (2025), a video diffusion model trained under the flow matching Lipman et al. (2022) framework. We extend its image-to-video capability to generate 6D videos as $V = \{X_t^{RGB},\ X_t^{XYZ}\}_{t=0}^{T-1}$. $V$ is first encoded into a latent space via a VAE encoder $\mathcal{E}$: $x_1 = \mathcal{E}(V)$, and interpolating with a noise latent $x_0 \sim \mathcal{N}(0, I)$:

$$x_t = (1-t)x_0 + tx_1, \quad t \sim \mathcal{U}(0,1).\qquad(4)$$

And a velocity predictor $u$ is trained to regress the velocity between endpoints:

$$\mathcal{L}_{\text{FM}} = \mathbb{E}\left[\|u(x_t, c_{\text{img}}, c_{\text{txt}}, t) - (x_1 - x_0)\|^2\right],\qquad(5)$$

where $c_{\text{img}}$ and $c_{\text{txt}}$ denote the image and text condition embeddings. This formulation enables efficient learning of temporally coherent and geometrically consistent 6D video sequences.

## 4.2  FUSION STRATEGIES

To finetune the video diffusion model for joint RGB and XYZ generation, a key challenge is designing an effective fusion strategy that enables the model to leverage both modalities. Our goal is to exploit the strong priors of pretrained models through simple yet effective fusion designs. Motivated by

Figure 4: **Overview of 4DNeX.** Given a single RGB image and an initialized XYZ map, 4DNeX encodes both inputs with a VAE encoder and fuses them via width-wise concatenation. The fused latent, combined with a noise latent and a guided mask, is processed by a LoRA-tuned Wan-DiT model to jointly generate RGB and XYZ videos. A lightweight post-optimization step recovers camera parameters and depth maps from the predicted outputs.

prior work, latent concatenation is a widely adopted technique for joint modeling. We systematically explore fusion strategies across different dimensions, as illustrated in Fig. B.

**Channel-wise Fusion.** A straightforward approach is to concatenate RGB and XYZ along the channel dimension, and insert a linear layer (*a.i*) or a modality switcher (*a.ii*) to adapt the input and output formats. However, this strategy disrupts the input and output distributions expected by the pretrained model, which undermines the benefits of pretraining. It typically requires large-scale data and substantial computational resources to achieve satisfactory performance.

**Batch-wise Fusion.** To maintain pretrained distributions, this strategy treats RGB and XYZ as separate samples and uses a switcher to control the output modality (*b.i*). While it preserves unimodal performance, it fails to establish cross-modal alignment. Even with additional cross-domain attention layers (*b.ii*), the modalities remain poorly correlated.

**Frame-/Height-/Width-wise Fusion.** These strategies concatenate RGB and XYZ along the frame (*c*), height (*d*), or width (*e*) dimensions, preserving the distributions of the pretrained model while enabling cross-modal interaction within a single sample. We analyze them from the perspective of token interaction distance. Intuitively, shorter interaction distance between corresponding tokens makes it easier for the model to learn cross-modal alignment. As shown in Fig. 3, width-wise fusion yields the shortest interaction distance, leading to more effective alignment and higher generation quality, as confirmed by our experiments (Sec. 5.3).

## 4.3 NETWORK ARCHITECTURE

As illustrated in Fig. 4, our framework takes a single image $I_0 \in \mathbb{R}^{H \times W \times 3}$ and an initialized XYZ map $X^{init} \in \mathbb{R}^{H \times W \times 3}$ as conditions. Both are encoded by a frozen VAE encoder and concatenated along the width dimension. This fused condition is then combined with a noise latent $x_t$ and a binary mask $M$ along the channel dimension, and fed into a pretrained DiT with LoRA tuning. The output latent is decoded by a VAE decoder to generate paired RGB and XYZ video sequences. A lightweight post-optimization step further recovers camera parameters and depth maps from the predicted outputs.

**XYZ Initialization.** We initialize the first-frame XYZ map $X^{init}$ using a sloped depth plane. Specifically, we define a normalized 2D coordinate grid over the range $[-1, 1]^2$ and compute the initial XYZ values as:

$$X^{init}_{i,j} = \left( \frac{2j}{W-1} - 1, \ \frac{2i}{H-1} - 1, \ \frac{2i}{H-1} - 1 \right). \tag{6}$$

This results in a sloped plane where depth values gradually increase from the bottom to the top of the image, reflecting common depth priors in natural scenes (*e.g.*, sky regions appearing farther away). Such initialization provides a stable starting point for geometry learning.

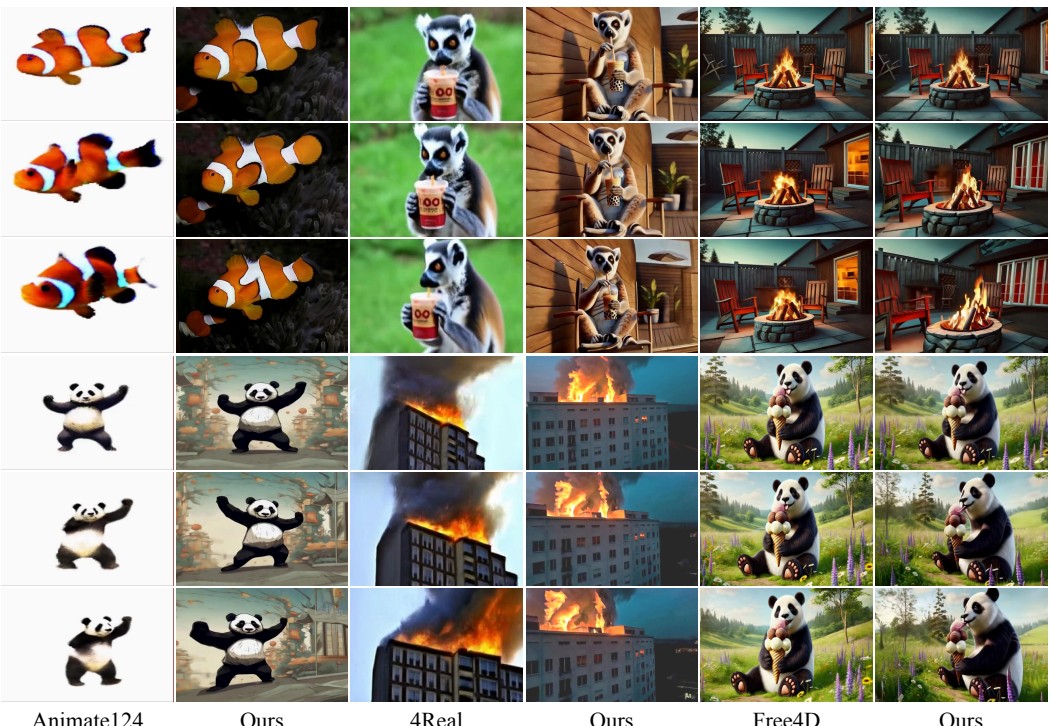

| Animate124 | Ours | 4Real | Ours | Free4D | Ours |

Figure 5: **Qualitative comparison.** Our method generates results with higher consistency, better aesthetics, and notably larger motion than existing 4D generation methods.

**XYZ Normalization.** Since the VAE is pretrained on RGB images, directly encoding XYZ inputs with different distributions can cause instability and suboptimal performance. To mitigate this issue, inspired by Chen et al. (2025) , we apply a modality-aware normalization strategy to adapt the XYZ latent to the pretrained VAE's distributional priors. Specifically, we compute the mean $\mu$ and standard deviation $\sigma$ of XYZ latent across the training dataset, and normalize the encoded representation as: $\hat{x} = \frac{x-\mu}{\sigma}$, where $x$ denotes the XYZ latent. Before passing into the VAE decoder, we perform de-normalization to recover the original scale: $x = \hat{x} \cdot \sigma + \mu$.

**Mask Design.** Following Wan et al. (2025), we introduce a guided mask $M \in [0,1]^{T \times H \times W}$, where $M_{t,i,j} = 1$ indicates a known pixel and $M_{t,i,j} = 0$ indicates a pixel to be generated. Since we use an approximate initialization for the first-frame XYZ map, we assign a soft mask:

$$M_{0,i,j}^{XYZ} = 0.5, \quad \forall i, j, \tag{7}$$

which encourages the model to refine the initial geometry during generation.

**Modality-Aware Token Encoding.** To preserve pixel-wise alignment across modalities during joint modeling, we adopt a shared rotary positional encoding (RoPE) Su et al. (2024) for RGB and XYZ tokens. To further distinguish their semantic differences, we introduce a learnable domain embedding. Given RGB and XYZ token sequences $x^{\text{RGB}}, x^{\text{XYZ}} \in \mathbb{R}^{L \times D}$, we apply the following encoding:

$$
\begin{aligned}
x^{RGB} &\leftarrow \text{RoPE}(x^{RGB}) + e_{RGB}, \\
x^{XYZ} &\leftarrow \text{RoPE}(x^{XYZ}) + e_{XYZ},
\end{aligned}
\tag{8}
$$

where $\text{RoPE}(\cdot)$ denotes the shared rotary positional encoding, and $e_{RGB}, e_{XYZ} \in \mathbb{R}^{1 \times D}$ are learnable domain embeddings broadcasted across the sequence.

**Post-Optimization.** Since our method produces XYZ videos that represent dense 3D points in global coordinates, we can recover the corresponding camera parameters $C = (R, t, K)$ and depth maps $d$ for the generated RGB frames via a lightweight post-optimization step. Specifically, we minimize the reprojection error between the generated and back-projected 3D coordinates:

$$\min_{R,t,K,d} \sum_{i,j} \left\| \tilde{q}_{i,j}^{XYZ} - \hat{q}_{i,j}^{XYZ} \right\|_2^2, \tag{9}$$

Table 1: **4D Generation Results on VBench Huang et al. (2024).** We report the consistency, dynamics, and aesthetics of the generated videos, together with the inference time of each method.

| Method | Consistency ↑ | Dynamic ↑ | Aesthetic ↑ | Time (min) ↓ |
|---|---|---|---|---|
| 4Real Yu et al. (2024a) | 95.7% | 32.3% | 50.9% | 90 |
| Free4D Liu et al. (2025) | 96.0% | 47.4% | 64.7% | 60 |
| Ours | 96.4% | 58.0% | 59.5% | 15 |
| Animate124 Zhao et al. (2023) | 90.7% | 45.4% | 42.3% | \ |
| Free4D Liu et al. (2025) | 96.9% | 40.1% | 60.5% | 60 |
| Ours | 97.2% | 58.3% | 53.0% | 15 |
| GenXD Zhao et al. (2024) | 89.8% | 98.3% | 38.0% | \ |
| Free4D Liu et al. (2025) | 96.8% | 100.0% | 57.9% | 60 |
| Ours | 96.8% | 100.0% | 52.4% | 15 |

Table 2: **User study results.** Percentages indicate user preference.

| Comparison | Consistency | Dynamic | Aesthetic |
|---|---|---|---|
| Ours vs. Free4D Liu et al. (2025) | 56% / 44% | 59% / 41% | 53% / 47% |
| Ours vs. 4Real Yu et al. (2024a) | 79% / 21% | 85% / 15% | 93% / 7% |
| Ours vs. Animate124 Zhao et al. (2023) | 75% / 25% | 56% / 44% | 100% / 0% |
| Ours vs. GenXD Zhao et al. (2024) | 90% / 10% | 85% / 15% | 100% / 0% |

where $\hat{q}_{i,j}^{XYZ}$ denotes the generated 3D coordinate, and $\tilde{q}_{i,j}^{XYZ}$ is computed by back-projecting the depth value into 3D space:

$$\tilde{q}_{i,j}^{XYZ} = [R \mid t]^{-1} K^{-1} \left( d_{i,j} \cdot [i, j, 1]^{\top} \right). \tag{10}$$

This optimization is computationally efficient and can be parallelized across views, producing physically plausible and geometrically consistent estimates of camera poses and depth maps.

## 5 EXPERIMENTS

### 5.1 SETTING

**Baselines.** Following Liu et al. (2025), we compare against both text-to-4D and image-to-4D approaches. For text-to-4D, we evaluate 4Real Yu et al. (2024a), a state-of-the-art method. For image-to-4D, we benchmark Free4D Liu et al. (2025), GenXD Zhao et al. (2024), and the object-level method Animate124 Zhao et al. (2023). For text-to-4D, we first generate an image from the prompt and then apply the same image-to-4D pipeline. To ensure fairness, all methods use the same single image or prompt during evaluation.

**Datasets and Metrics.** We conduct evaluations using images and texts sourced from the official project pages of the compared methods. To assess the quality of generated novel-view videos, We report standard VBench metrics Huang et al. (2024), including Consistency (averaged over subject and background), Dynamic Degree, and Aesthetic Score. Given the lack of a well-established benchmark for 4D generation, we further conduct a user study involving 23 evaluators to enhance the reliability of our evaluation.

**Implementation Details.** We adopt the Wan2.1 Wan et al. (2025) image-to-video model (14B parameters) as our base. For the modality-aware normalization, we normalize XYZ latents with fixed statistics estimated from the training set. To efficiently adapt the base model to image-to-4D generation, we apply LoRA finetuning (rank 64) rather than full-parameter finetuning. Novel-view videos are obtained by generating 4D point clouds with our model and rendering them with TrajectoryCrafter YU et al. (2025). Full training configurations are provided in the Appendix.

### 5.2 MAIN RESULTS

**4D Geometry Generation.** As illustrated in Fig. F, we visualize the paired RGB and XYZ video generated from a single image. The results demonstrate that our method can simultaneously infer

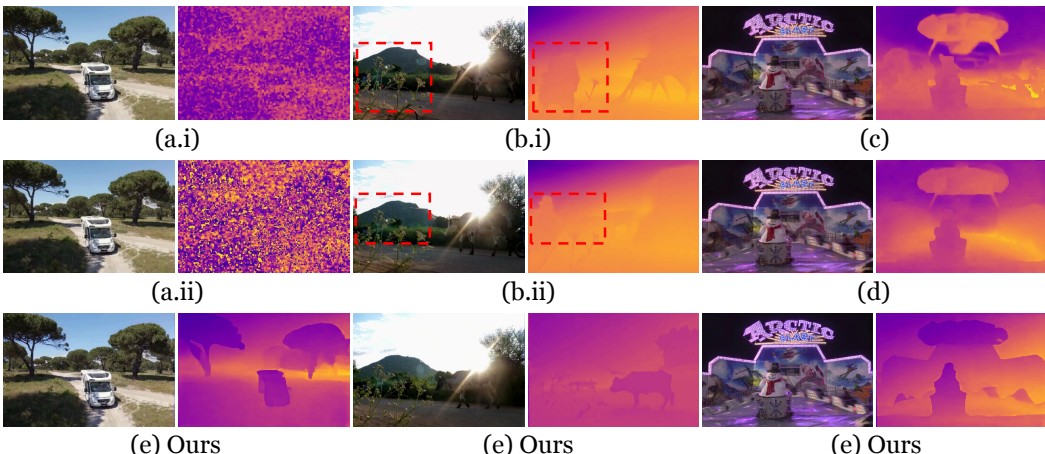

(a.i)           (b.i)           (c)

(a.ii)           (b.ii)           (d)

(e) Ours           (e) Ours           (e) Ours

Figure 6: **Ablation study on fusion strategies.** We compare channel-wise (a), batch-wise (b), frame-wise (c), height-wise (d), and our width-wise fusion (e) for RGB and XYZ inputs.

plausible scene motion and the corresponding 4D geometry from a single image. This high-quality geometric representation of dynamic scenes is essential for consistent and photorealistic novel view synthesis in the subsequent rendering stage.

**Novel-View Video Generation.** Quantitative results (Table 1) show our method matches state-of-the-art performance and surpasses others on Dynamic Degree. Free4D Liu et al. (2025) gains higher aesthetic scores from the proprietary Kling Team (2024) model. Qualitative results (Fig. 5) further highlight our stronger and more coherent dynamics under camera motion. User study results (Table 2) confirm consistent preference for our method in consistency, dynamics, and aesthetics. While performance is close to Free4D, its test set is object-centric, whereas our approach generalizes to diverse in-the-wild scenarios (Fig. E). Moreover, our feed-forward design enables efficient generation (∼15 min), compared to over one hour for Free4D's multi-stage pipeline.

### 5.3 ABLATIONS AND ANALYSIS

To validate the effectiveness of our used width-wise fusion strategy and support the analysis presented in Sec. 4.2, we conduct an ablation study comparing five different fusion designs, as illustrated in Fig.6. Among these, channel-wise fusion introduces a severe distribution mismatch with the pretrained prior, often leading to noisy or failed predictions (*a.i-a.ii*). Batch-wise fusion preserves unimodal quality but fails to capture cross-modal alignment, yielding inconsistent RGB-XYZ correlation (*b.i-b.ii*). Frame-wise (*c*) and height-wise (*d*) strategies provide moderate improvements, yet still suffer from suboptimal alignment and visual quality. In contrast, our width-wise fusion brings corresponding RGB and XYZ tokens closer in the sequence, significantly shortening the cross-modal interaction distance. This facilitates more effective alignment and yields sharper, more consistent geometry and appearance across frames, as demonstrated in Fig. 6 (e).

### 6 CONCLUSION

We present 4DNeX, the first feed-forward framework for generating 4D scene representations from a single image. Our approach fine-tunes a pretrained video diffusion model to enable efficient image-to-4D generation. To address the scarcity of training data, we construct 4DNeX-10M, a large-scale dataset with high-quality pseudo-4D annotations. Furthermore, we propose a unified 6D video representation that jointly models appearance and geometry, along with a set of simple yet effective adaptation strategies to repurpose video diffusion models for the 4D generation task. Extensive experiments demonstrate that 4DNeX generates high-quality dynamic point clouds, providing a reliable geometric foundation for synthesizing novel-view videos. The resulting videos achieve competitive performance compared to existing methods, while offering superior efficiency and generalizability. We hope this work paves the way for scalable and accessible single-image generative 4D world modeling.

## ETHICS STATEMENT

This work does not involve human subjects, personally identifiable information, or sensitive data. The datasets used in this study are publicly available and widely adopted in the machine learning community. All experiments were conducted using standard computational resources without environmental or societal harm. The methodology does not introduce discriminatory biases, and the model's potential applications are aligned with responsible AI principles. The authors have reviewed the ICLR Code of Ethics and confirm that this submission adheres to its guidelines.

## REPRODUCIBILITY STATEMENT

To support reproducibility, we provide a complete description of our model architecture, training procedures, hyperparameters, and evaluation protocols in the main paper. Additional implementation details, including data preprocessing steps and environment specifications, are included in the Appendix. We have strived to document all necessary components with sufficient clarity to enable independent replication of our results. Beyond documentation, we are committed to publicly releasing our codebase, pretrained model weights, and accompanying resources. This commitment reflects our broader objective of fostering transparency, promoting open scientific exchange, and empowering the research community to reproduce, scrutinize, and extend our findings in future investigations.

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

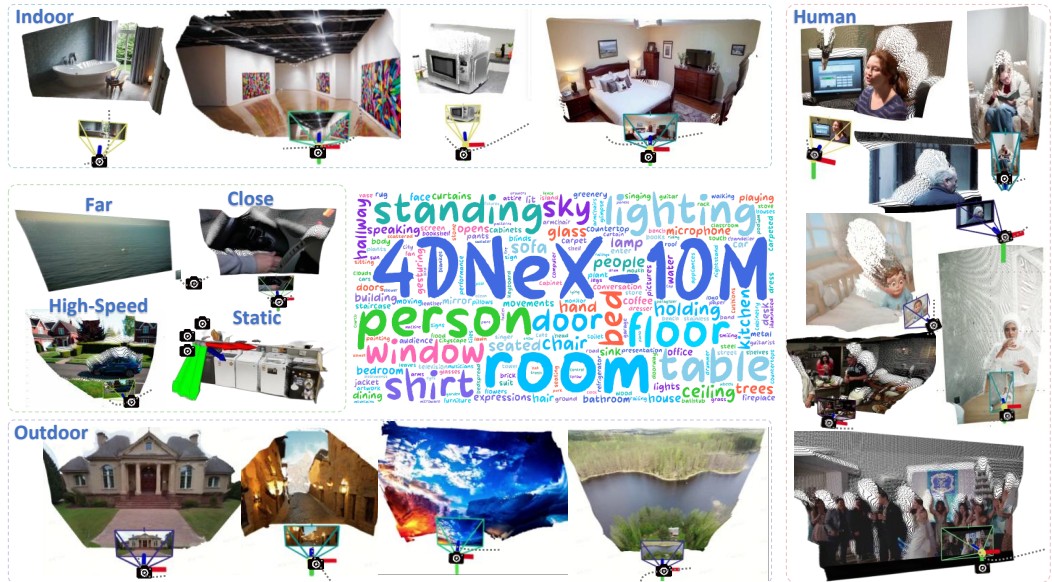

Figure A: **Visualization of 4DNeX-10M Dataset.** Our dataset spans a wide range of dynamic scenarios, including indoor, outdoor, close-range, far-range, static, high-speed, and human-centric scenes. The word cloud summarizes common visual concepts captured in the dataset, while the 4D point clouds and camera trajectories demonstrate the spatial precision of our pseudo-annotations.

## LLM USAGE

We acknowledge large language models (LLMs) in the preparation of this manuscript. Specifically, we utilized LLMs for text polishing, grammar correction, and improving the clarity. The core experimental results and scientific contributions remain entirely our own work.

## A   MORE IMPLEMENTATION DETAILS

We opt for the vanilla Wan2.1 Wan et al. (2025) image-to-video model as our final base model with a total of 14B parameters[1]. Most importantly, given the significant distribution gap between the spatial coordinates XYZ and the original RGB domain, one may carefully deal with the normalization of the input data to the diffusion model so that the noise scheduling is balanced across two modalities. Recall our diffusion target is jointly denoising RGB and XYZ where the noised RGB latent is in the space of KL-regularized VAE whose distribution is close to a Gaussian Distribution. However, the XYZ coordiante is not normally distributed in the 3D space, which leads to modality gap during denoising. To bridge this gap, we propose to perform modality-aware normalization. Specifically, we trace the statistics (mean and standard deviation) of XYZ domain in the latent space over 5K random samples from the training dataset. It results in $\mu = -0.13$ and $\sigma = 1.70$, which serves as the constant normalization term for XYZ latent during training and inference. To fully transfer the capability of original image-to-video generation from the base model to the target image-to-4D task, we train a LoRA with a rank of 64 for the sake of parameter and data efficiency instead of full-parameter supervised finetuning. The Lora finetuning is run with a batch size of 32 using an AdamW optimizer. The learning rate is set to $1 \times 10^{-4}$ with a cosine learning rate warmup. The training is distributed on 32 NVIDIA A100 GPUs with 5k iterations at a spatial resolution of $480 \times 720$ for each modality. To generate novel-view videos, we first produce a 4D point cloud representation of the scene using our feed-forward model, and then render the results using YU et al. (2025).

**Cross-Domain Self-Attention (CDSA)** As introduced in Sec. 4.2, we introduce a Cross-Domain Self-Attention (CDSA) module to enhance the alignment between RGB and XYZ modalities, particularly under the batch-wise fusion strategy. Figure D illustrates the architecture of this module.

---

[1] https://huggingface.co/Wan-AI/Wan2.1-I2V-14B-480P

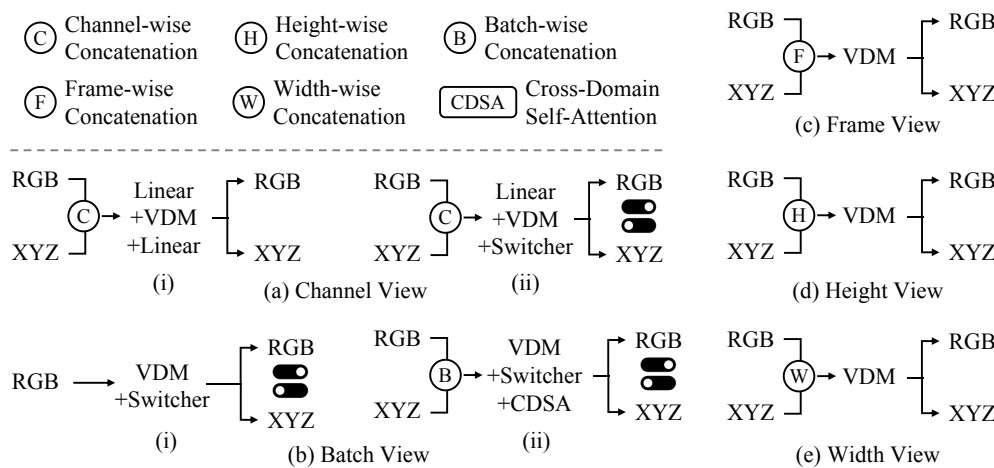

Figure B: **Comparison of fusion strategies for joint RGB and XYZ modeling.** We explore five fusion strategies and analyze their impact on model compatibility and cross-modal alignment.

As shown in the left part of Fig. D, the CDSA block is inserted between the standard self-attention and cross-attention layers within a transformer block. It explicitly enables bidirectional interaction between RGB and XYZ tokens through attention mechanisms—allowing RGB tokens to attend to XYZ tokens and vice versa—thus facilitating cross-modal information exchange.

To balance performance and efficiency, we implement and compare two versions of CDSA:

- Full Version: All RGB and XYZ tokens participate in dense cross-domain attention. This version achieves stronger modality interaction at the cost of higher memory and computation.

- Sparse Version: Token interactions are restricted to spatially corresponding positions between RGB and XYZ sequences. This reduces overhead while retaining most of the alignment benefits.

While both versions aim to bridge the modality gap by promoting fine-grained token-level communication, our experiments reveal that under the batch-wise fusion setting (Fig. B (b.ii)), even with CDSA, the overall cross-modal alignment remains limited. This is primarily due to the spatial separation of RGB and XYZ tokens, which contrasts with the more effective width-wise fusion strategy (Fig. B (e)) where the interaction distance is inherently shorter.

## B  DETAILS OF USER STUDY

**User Study: Comparison with Existing Methods.** To evaluate the effectiveness of our method, we conducted a user study comparing it against several existing approaches. We collected a total of 74 video pairs, each generated from the same input image or text prompt to ensure fair comparisons. Competing methods included Free4D Liu et al. (2025), 4Real Yu et al. (2024a), GenXD Zhao et al. (2024), and Animate124 Zhao et al. (2023). All comparison videos were obtained from their official project pages. The study was conducted online, and a screenshot of the evaluation interface is shown in Fig. C. Participants were asked to assess each video pair across three criteria: Consistency, Dynamics, and Aesthetics. For each criterion, they were instructed to choose the video they perceived as better. If a comparison was too difficult to judge, they could skip to the next example without selecting an answer. All responses were collected anonymously, and no personal data were recorded during the study.

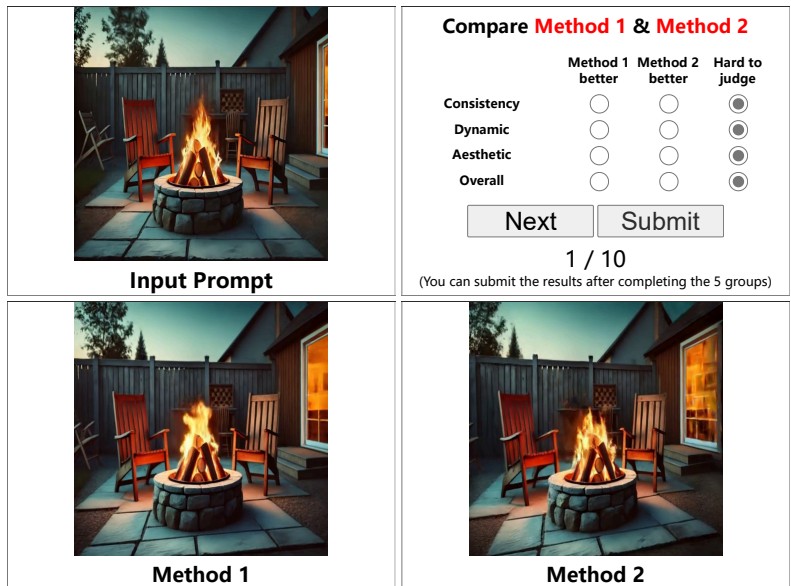

Figure C: **User study interface.** Participants were shown an input prompt and two generated videos from different methods. They were asked to compare the results based on *Consistency*, *Dynamics* and *Aesthetics*. Each question allowed skipping if the difference was hard to judge.

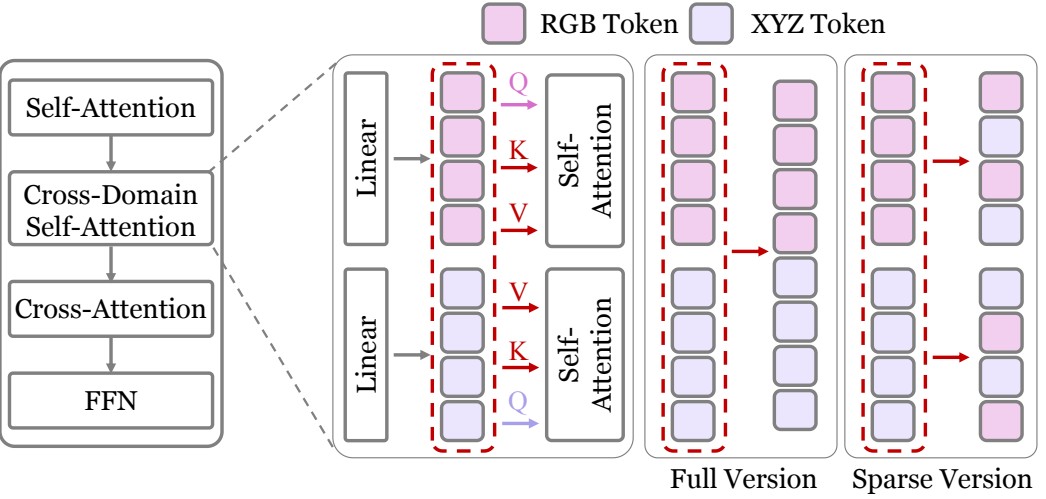

Figure D: **Architecture of the Cross-Domain Self-Attention (CDSA) module.** The CDSA block is inserted between self-attention and cross-attention layers to facilitate bidirectional interaction between RGB and XYZ modalities. We explore two variants: the *Full Version*, where all tokens interact densely, and the *Sparse Version*, where attention is restricted to spatially corresponding token pairs. This design enables effective cross-modal alignment with different trade-offs in efficiency and performance.

## C  DETAILS OF VBENCH METRICS

To comprehensively evaluate the quality of our synthesized novel-view videos, we adopt a suite of metrics introduced in VBench Huang et al. (2024), covering three key aspects: *Consistency* (for both subject and background), *Degree of Motion*, and *Aesthetic Quality*.

**Subject / Background Consistency.** This metric assesses how consistently both the main subject (*e.g.*, human, vehicle, animal) and the surrounding background are maintained throughout the video.

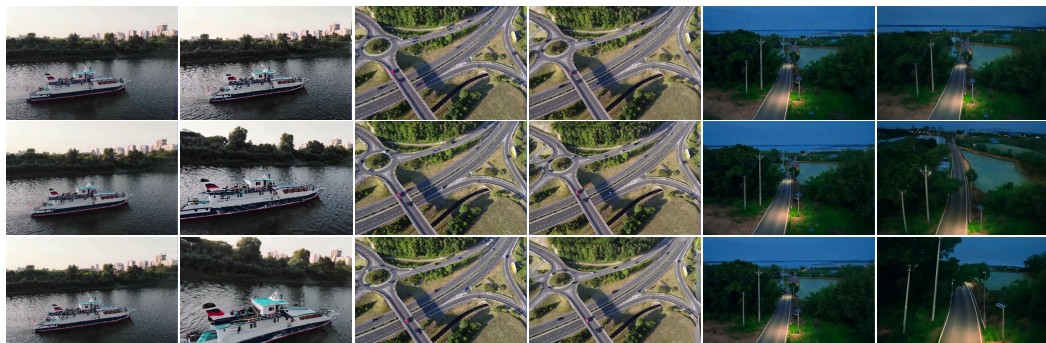

Generated Video    Novel-view Video    Generated Video    Novel-view Video    Generated Video    Novel-view Video

Figure E: **Novel-view video results on in-the-wild data.**

It leverages feature similarity across frames using DINO Caron et al. (2021) for the foreground and CLIP Radford et al. (2021) for the background. DINO focuses on preserving subject identity by comparing learned visual representations, while CLIP captures broader scene coherence. The average of both provides a balanced view of overall temporal consistency.

**Degree of Motion.** To avoid favoring overly static videos that may perform well on consistency metrics, we include a motion-aware measure. Specifically, RAFT Teed & Deng (2021) is applied to estimate optical flow, and the *Dynamic Degree* is computed by averaging the top $5\%$ of largest flow magnitudes. This helps emphasize prominent movements, such as object actions or camera shifts, while de-emphasizing negligible or noisy motions, ensuring a more meaningful evaluation of dynamics.

**Aesthetic Quality.** To reflect the perceived visual appeal of the generated videos, we utilize the LAION Aesthetic Predictor LAION-AI (2022), a lightweight regressor trained atop CLIP features to score image aesthetics on a scale from $1$ to $10$. It considers multiple factors, including color composition, realism, layout, and overall artistic impression. We apply this predictor to each frame and report the average score as the final *Aesthetic Quality* metric.

**Limitations and Future Work** While 4DNeX demonstrates promising results in single-image 4D generation, several limitations remain. First, our method relies on pseudo-4D annotations for supervision, which may introduce noise or inconsistencies, particularly in fine-grained geometry or long-term temporal coherence. Introducing high-quality real-world or synthetic dataset would be fruitful for general 4D modeling. Second, although the image-driven generated results are 4D-grounded, controllabilities over lighting, fine-grained motion and physical property are still lacking. Third, the unified 6D representation, while effective, assumes relatively clean input images and may degrade under occlusions, extreme lighting conditions, or cluttered backgrounds. Future work includes improving temporal modeling with explicit world priors, incorporating real-world 4D ground-truth data when available, and extending our framework to handle multi-object or interactive scenes. Additionally, integrating multi-modal inputs such as text or audio could further enhance controllability and scene diversity.

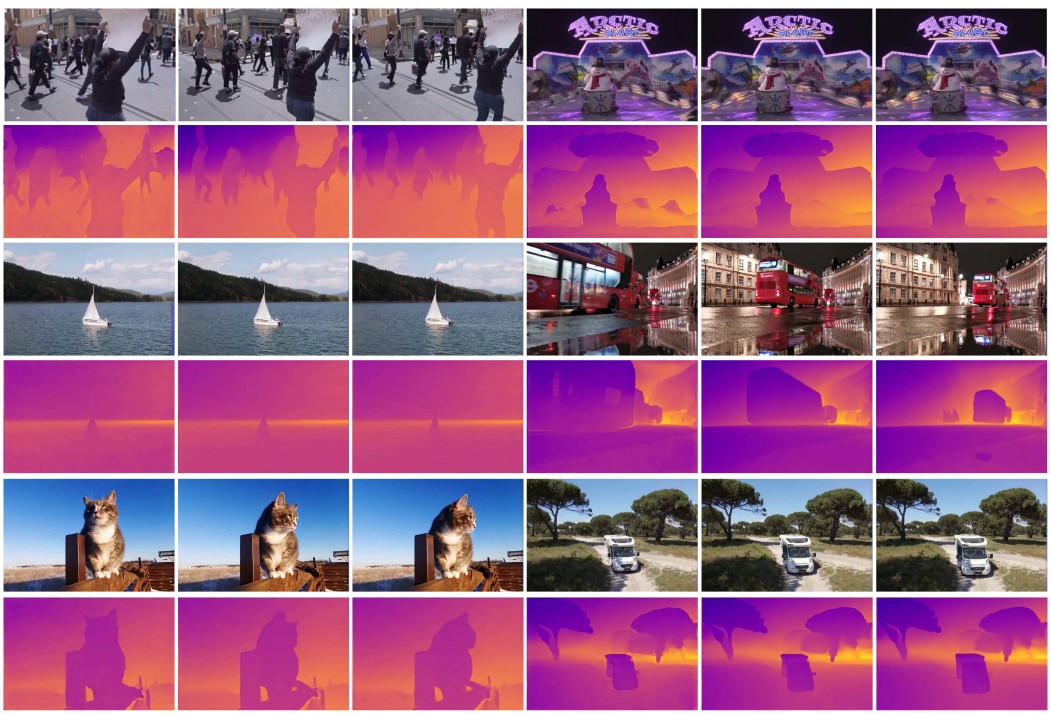

Figure F: **Generated RGB and XYZ sequences from single-image input.** Each pair of rows shows the output RGB video and its corresponding XYZ sequence.

