# OpenReview forum: "4DNeX: Feed-Forward 4D Generative Modeling Made Easy"
_ICLR.cc/2026/Conference — ICLR 2026 Conference Withdrawn Submission_

### Official Review · Reviewer_sm7K · 2025-10-26

**Soundness:** 3
**Presentation:** 3
**Contribution:** 3
**Rating:** 6
**Confidence:** 3

**Summary:**

4DNeX is a feed-forward framework that generates dynamic 4D scene representations (RGB + XYZ sequences) directly from a single image. It reformulates 4D generation as 6D video diffusion, fine-tuning a 14B Wan2.1 video diffusion model via a lightweight width-wise fusion and modality-aware normalization strategy. It introduces 4DNeX-10M, a large-scale pseudo-annotated dataset curated from DUSt3R / MonST3R / MegaSaM pipelines, and performs minimal post-optimization to recover camera poses. The method achieves competitive or superior motion quality to Free4D / 4Real / Animate124 while being significantly faster (~15 min vs 1+ hr), positioning itself as a scalable path toward single-image 4D world modeling.

**Strengths:**

1. “RGB + XYZ as 6D video” is conceptually elegant — unifies appearance & geometry without NeRF volume rendering or Gaussian splats during training.

2. Width-wise fusion is empirically validated and actually justified with token interaction distance。

3. LoRA-only tuning on 14B Wan2.1 while preserving pretrained RGB distributions is a strong practical insight, compared to Cat4D / Free4D which lose RGB appearance fidelity when optimizing all parameters.

4. Post-optimization for camera recovery is lightweight & well-justified.

**Weaknesses:**

1. Core novelty is incremental, effectively “fine-tuning Wan2.1 to predict XYZ instead of RGB” with latent concatenation and normalization.
There is no fundamentally new 4D generative architecture. This feels closer to a careful repurposing of a large pretrained model rather than a new generative paradigm.

2. No true camera control or 3D consistency is learned during generation itself. The “feed-forward” claim is slightly misleading — XYZ is predicted in image-plane coordinates, not explicitly SE(3)-aware.

3. The point cloud visualization in the video shows that the geometry of the generated scenes is unnatural.

**Questions:**

1. How to guarantee that the generated XYZ is geometrically accurate? Please add quantitative evaluations of 3D error or global pose alignment.

2. Is your method actually SE(3)-consistent, or does the 4D break when camera motion is large? Please show failure cases with fast curved trajectories, not only forward panning.

3. The sloped plane (Eq. 6) is a very strong inductive bias. Do you ablate training without any depth init? Does the model still converge?

---

### Official Review · Reviewer_gKPd · 2025-11-01

**Soundness:** 3
**Presentation:** 3
**Contribution:** 2
**Rating:** 4
**Confidence:** 4

**Summary:**

The paper introduces 4DNeX, which is proposed as the first feed-forward framework for generating 4D (dynamic 3D) scene representations from a single image. In contrast to prior optimization-based methods, this approach offers an efficient, end-to-end image-to-4D generation pipeline by fine-tuning a pretrained video diffusion model. The final output is a 6D video (RGB + XYZ sequences) which can be converted into dynamic point clouds and subsequently used for novel-view video synthesis

**Strengths:**

1. The authors introduce a successful strategy for adapting existing, powerful video diffusion models to the 4D domain. The systematic investigation into fusion mechanisms, culminating in the adoption of width-wise fusion, is an important technical finding for jointly modeling appearance and geometry sequences
2. 4DNeX is the first feed-forward method to tackle the challenging task of single-image-to-4D generation. This feed-forward nature makes it significantly more efficient than computationally intensive optimization-based approaches, demonstrating a generation time of about 15 minutes compared to 60-90 minutes for competitors

**Weaknesses:**

1. The fundamental technical contribution is primarily an engineering adaptation of a pretrained video generation model to a new input/output domain. As the reviewer notes, the RGB-XYZ representation itself is not new (Zhang et al.). The novelty lies in the generation paradigm (image-to-4D, feed-forward), but the algorithmic advancements beyond the fine-tuning strategies are limited, which may lead to a low technical score
2. The large-scale training is entirely dependent on pseudo-4D annotations generated by external reconstruction models (MonST3R, MegaSaM). The authors acknowledge that this process "may introduce noise or inconsistencies, particularly in fine-grained geometry or long-term temporal coherence," which fundamentally limits the quality ceiling of the trained model
3. The image-to-4D ability comes from Wan-I2V. Why not considering the pipeline that first generating a video then conducting 4D reconstruction? This pipeline could also get dynamic point cloud  as output.

**Questions:**

1. How to get novel view rendering results from dynamic point cloud?
2. See weakness

---

### Official Review · Reviewer_uV7i · 2025-11-01

**Soundness:** 2
**Presentation:** 2
**Contribution:** 2
**Rating:** 4
**Confidence:** 4

**Summary:**

The paper proposes a feed-forward framework for image-to-4D generation, adapting a video diffusion model to jointly predict RGB frames and corresponding XYZ point cloud coordinates. For fine-tuning the video diffusion model, the authors construct a large-scale 4D dataset by employing feed-forward 4D reconstruction models to provide 3D/4D annotations for in-the-wild videos. Several spatial fusion strategies for combining image and geometry modalities are explored, with concatenation along the width axis achieving the best performance. A post-processing optimization step is introduced for camera pose estimation from the generated video and for producing a global point cloud. Experiments conducted on both public and collected datasets demonstrate the effectiveness of the proposed approach.

**Strengths:**

1）The method is conceptually simple and easy to follow.

2）The work targets a feed-forward model for image-to-4D generation, which is a practical and relevant research topic.

3）Experimental results show that the proposed method achieves state-of-the-art performance in its evaluated settings.

**Weaknesses:**

1） The paper does not mention Aether, which, to my knowledge, is the first work that leverages video diffusion models for joint RGB and geometry (camera ray maps and depth maps) prediction, and also supports image-conditioned 4D generation, albeit as a world model. The authors should discuss the differences, advantages, and disadvantages relative to Aether, and, if possible, include comparison results to strengthen the work.

2） For annotation, the authors rely solely on feed-forward models (Monst3R and MegaSAM) to provide pseudo 3D/4D labels for in-the-wild videos, which are then used to fine-tune the video diffusion model. This approach may degrade RGB generation quality during fine-tuning, and, although the method targets 4D geometry, the annotation pipeline itself uses fast feed-forward geometry estimation from video. An alternative worth exploring is to first perform high-quality image-to-video generation using state-of-the-art video generation models (e.g., Wan, Keling), and then reconstruct the generated videos using strong optimization-based reconstruction methods (e.g., PIE3). This could leverage existing high-quality modules “for free” and potentially outperform geometry prediction via video diffusion in efficiency. The authors are encouraged to discuss this possibility and, if feasible, add experiments to compare the two pipelines.

3）The current comparisons are not entirely fair: prior methods such as Animate124, 4Real, and Free4D generate explicit 4D representations (e.g., 4D Gaussian Splatting or dynamic NeRF) that support free-view camera control and novel-view rendering, while the proposed method only produces RGB frames and point clouds, lacking camera pose control and novel-view capability. Comparing novel-view renderings from prior methods with generated videos from TrajectoryMaster is thus unequal. The authors are suggested to apply post-optimization techniques, as in previous works, to build a 4DGS representation from their predicted camera poses, and then perform comparisons using 4DGS renderings.

4）The paper does not provide geometry metric evaluation (e.g., point cloud accuracy, completeness, or consistency metrics) for the generated 4D outputs. Such evaluation is essential to objectively measure the quality of geometry prediction and would strengthen the experimental analysis. The authors should design and report appropriate geometry evaluation metrics to support their claims.

**Questions:**

Is there any generation quality degrade for rgb video after finetuning？

---

### Official Review · Reviewer_QGiz · 2025-11-01

**Soundness:** 3
**Presentation:** 3
**Contribution:** 2
**Rating:** 4
**Confidence:** 4

**Summary:**

This paper introduces 4DNeX, presented as the first feed-forward framework for generating 4D (dynamic 3D) scene representations from a single image. It targets efficient image-to-4D generation by fine-tuning a pretrained video diffusion model (Wan2.1) to produce 6D videos (RGB + XYZ sequences). The authors build 4DNeX-10M, a large-scale dataset with over 9.2M annotated frames, and propose adaptation strategies including width-wise fusion, XYZ normalization, and post-optimization. Experiments on novel-view video synthesis show improved dynamics and efficiency compared to existing 4D generation methods.

**Strengths:**

- The paper tackles an important and challenging problem - generating 4D point clouds from a single image with pretrained video diffusion model.
- The construction of a dataset with 9.2M+ frames from diverse sources (DL3DV, RE10K, Pexels, Vimeo, VDM) addresses the scarcity of 4D training data.
- 15-minute generation vs >1 hour for Free4D demonstrates practical advantages.
- The progression from motivation to method to results is logical and easy to follow.

**Weaknesses:**

- The dataset relies entirely on pseudo-annotations from MonST3R, MegaSaM, and DUSt3R, yet there is no quantitative validation of these annotations against DROID-SLAM[1] which is use in many recent works as ground truth for 3D dynamic scenes.
- Unclear 3D point-cloud (XYZ) quality; more XYZ visualizations and comparisons to other methods would help.
- Accuracy of the post-optimized camera parameters is not analyzed.
- User study details are missing (e.g., number of participants).
- It’s unclear whether all demo video examples are outside the training and validation sets.
- No limitations section discussing failure modes or data bias.
- I’m not sure that directly concatenating RGB and XYZ latents should be called a new representation; however, the exploration of different fusion strategies is useful for future research.
- The dataset includes camera poses, but 4DNeX does not support camera-controlled generation. More discussion would be helpful. The current pipeline seems to allow arbitrary camera trajectories; adding camera-trajectory control would be more useful for real applications.
- For a fair comparison, GenXD should be trained on the same dataset (4DNeX-10M); otherwise, the comparison isn’t fair.
- I believe GEN3C also supports image-to-4D generation; please include a more complete comparison with GEN3C[2].
- 4DNeX uses a diffusion-based model (Wan2.1) as backbone, so I do not think it is accurate to say 4DNeX is the first feed-forward framework for 4D generation, because I think diffusion-based methods should not be called feed-forward methods.

[1] DROID-SLAM: Deep Visual SLAM for Monocular, Stereo, and RGB-D Cameras. In *Advances in Neural Information Processing Systems (NeurIPS)*, 2021.

[2] GEN3C: 3D-Informed World-Consistent Video Generation with Precise Camera Control. In *Proceedings of the IEEE/CVF Conference on Computer Vision and Pattern Recognition (CVPR)*, 2025.

**Questions:**

- Using TrajectoryCrafter to validate NVS performance may be insufficient. As far as I know, TrajectoryCrafter (itself diffusion-based) can alter details of the input point cloud. Can you provide additional NVS analyses that do not rely on TrajectoryCrafter?
- The video VAE is not trained on XYZ maps. Do the XYZ latents remain meaningful, and can the VAE reconstruct XYZ maps adequately? Please provide more details.
- Please clarify how masks are generated. Since you operate in latent space rather than pixel space, how do you map a pixel-space mask to a latent-space mask?
- Please include an ablation on the post-optimization: how much improvement does it bring, and how stable is the optimization?
- Will the dataset and trained model/code be released?

---

### Note · Authors · 2025-11-13

I have read and agree with the venue's withdrawal policy on behalf of myself and my co-authors.